# A Simulation Framework of Unmanned Aerial Vehicles Route Planning Design and Validation for Landslide Monitoring

Dongmei Xie [1], Ruifeng Hu [1], Chisheng Wang [1,*], Chuanhua Zhu [1], Hui Xu [2,3] and Qipei Li [1]

[1] Ministry of Natural Resources (MNR) Key Laboratory for Geo-Environmental Monitoring of Great Bay Area & Guangdong Key Laboratory of Urban Informatics, School of Architecture & Urban Planning, Shenzhen University, Shenzhen 518060, China; xiedongmei2020@email.szu.edu.cn (D.X.); huruifeng2020@email.szu.edu.cn (R.H.); czhu@szu.edu.cn (C.Z.); 2020104005@email.szu.edu.cn (Q.L.)

[2] School of Fine Arts and Design, Shenzhen University, Shenzhen 518060, China; xuhui@szu.edu.cn

[3] State Key Laboratory of Subtropical Building and Urban Science, Shenzhen 510640, China

*   Correspondence: wangchisheng@szu.edu.cn

**Abstract:** Unmanned aerial vehicles (UAVs) have emerged as a highly efficient means of monitoring landslide-prone regions, given the growing concern for urban safety and the increasing occurrence of landslides. Designing optimal UAV flight routes is crucial for effective landslide monitoring. However, in real-world scenarios, the testing and validating of flight path planning algorithms incur high cost and safety concerns, making overall flight operations challenging. Therefore, this paper proposes the use of the Unreal Engine simulation framework to design UAV flight path planning specifically for landslide monitoring. It aims to validate the authenticity of the simulated flight paths and the correctness of the algorithms. Under the proposed simulation framework, we then test a novel flight path planning algorithm. The simulation results demonstrate that the model reconstruction obtained using the novel flight path algorithm exhibits more detailed textures, with a 3D model simulation accuracy ranging from 10 to 14 cm. Among them, the RMSE value of the novel flight route algorithm falls within the range of 10 to 11 cm, exhibiting a 2 to 3 cm improvement in accuracy compared to the traditional flight path algorithm. Additionally, it effectively reduces the flight duration by 9.3% under the same flight path compared to conventional methods. The results confirm that the simulation framework developed in this paper meets the requirements for landslide damage monitoring and validates the feasibility and correctness of the UAV flight path planning algorithm.

**Keywords:** unmanned aerial vehicle; unreal engine; AirSim; landslide monitoring; flight path planning; simulation validation

## 1. Introduction

Landslides are one of the most prevalent geological hazards, involving the downward movement of large volumes of rocks, debris, or soil along slopes due to the force of gravity, heavy rainfall, earthquakes, changes in water levels, human activities, and so on [1,2]. Landslides of different slope gradients exhibit varying frequencies and velocities. They can manifest as various types, including flows, slides, topples, falls, or spreads, often exhibiting a combination of different movement patterns over time or during their lifecycle. In recent years, the occurrence frequency and scale of landslides have been increasing globally; this has been attributed to climate change and other contributing factors [3,4]. Landslides frequently result in extreme and destructive consequences, negatively impacting human safety, hindering urban planning, and causing significant economic losses [5–7]. The suddenness, complexity, and high uncertainty of landslides can present challenges in emergency monitoring, leading to delayed response, imprecise control, and inefficient disaster management. Consequently, monitoring and implementing emergency responses to landslide incidents have become focal points in the field of geological disaster prevention

and relief [8]. Therefore, the utilization of landslide monitoring technologies to gain insights into landslide behavior and reduce landslide risks is of paramount importance.

Traditional landslide monitoring relies on inclinometers to obtain precise measurements of slope magnitude and direction, while global navigation satellite systems (GNSSs) are utilized to monitor surface movement on the terrain. These instruments are capable of real-time monitoring and providing accurate data regarding landslide behavior, but they are limited in that each device can only generate measurement results for a single location [9]. In contrast, unmanned aerial vehicle (UAV) monitoring technology offers several advantages, such as real-time capabilities, high maneuverability, high-resolution imagery, and cost effectiveness [10], thus making UAVs more suitable for large-scale landslide monitoring. One crucial task that needs to be accomplished before conducting UAV monitoring is UAV flight path planning.

Efficient UAV flight path planning is useful for optimizing data acquisition and ensuring comprehensive coverage of the landslide area. UAVs can effectively capture high-resolution images of terrain if their flight paths are strategically designed, thus enabling precise monitoring and analysis of landslide behavior. There have been numerous comprehensive studies internationally on UAV flight path planning [11–14]. During the research process, extensive experimentation is essential, as unverified flight path algorithms can lead to errors. Therefore, testing and debugging are critical stages in UAV flight operations. Since algorithms are designed to operate on physical systems, real UAV testing is the most effective way to debug them [15]. Nevertheless, errors or incorrect parameter adjustments during testing can have severe consequences, such as triggering hazardous behaviors or even causing crashes. In the best-case scenario, this may only require repairing or rebuilding the UAV frame, but in the worst cases it could result in injuries to personnel. Furthermore, the challenging terrain in landslide areas presents significant obstacles for UAV flight operations, making on-site testing complicated, with a risk of UAV crashes and damage. Moreover, conducting UAV flights in urban areas may encounter legal restrictions, making algorithm validation in real scenarios both time consuming and costly [16]. These challenges emphasize the importance of developing a reliable and efficient UAV flight path planning method that can be tested and validated in a safe and controlled environment before actual implementation in real-life monitoring scenarios.

In response to the challenges and risks associated with conducting real-world UAV flight tests, many scholars have proposed using UAV simulators for flight simulations. Hentati et al. [17] introduced mainstream UAV simulation frameworks, such as XPlane [18], Flightgear [19], Gazebo [20], and JMavSim. However, these simulators have limitations in terms of assets and textures and lack support for motion capture (MOCAP). In contrast, simulators like AirSim and UE4 offer MOCAP support and enable the creation of UAV motion planning. Overall, JMavSim is relatively simple in terms of both configuration and functionality. XPlane and FlightGear are primarily designed for aircraft simulation and may not be well suited to the research of multirotor and fixed-wing unmanned aerial vehicles, limiting their available scenarios and applications. On the other hand, AirSim excels in these aspects. It provides dedicated interfaces that support the development and testing of deep learning and computer vision algorithms. Furthermore, it offers a wide variety of environments and scenarios, accommodating diverse research and testing requirements. Additionally, AirSim exhibits a high degree of scalability, allowing for the integration of various sensors and hardware devices. Table 1 gives the comparison between the proposed simulators. Matlekovic et al. [21] designed a microservices application for UAV infrastructure inspection, which involved high-level path planning, monitoring, and testing of autonomous UAV missions. They used Gazebo to simulate UAVs following waypoints provided by a cloud system to verify the correctness of the inspection path. While addressing communication and perception uncertainty, Ling et al. [22] proposed a scalable and maintainable distributed UAV swarm cooperative planning and perception–simulation framework. They used Bayesian reasoning-based estimation and distributed collaborative planning evaluation under incomplete perception to validate the framework's

working principle and practicality through simulation examples. Fernando et al. [23] utilized MATLAB®Simulink to simulate the model control algorithm and compared the quadrotor state model with the MATLAB®6DOF dynamic model in Aerospace Blockset, validating the model's effectiveness. However, this method could not run in real time after adding 3D visualization. Park et al. [24] proposed an improved real-time flight simulation method to predict the transient response of multi-rotor UAVs under gust influence. They combined unsteady rotor aerodynamics suitable for flight simulation with nonlinear flight dynamics to accurately predict UAV responses under gusts and verified the necessity of the simulation. While most of these simulation frameworks allow the creation of 3D worlds using different physics engines and sensor models, they still face challenges in creating large-scale complex scenes. Some frameworks, like Hector [25], lack support for popular hardware frameworks like Pixhawk and protocols like MavLink, limiting their usability. Moreover, the development and refinement of UAV flight path planning algorithms in the real world can be costly and involve significant risks. The use of UAV simulators offers a safer and more cost-effective alternative for testing and validating UAV flight path planning methods before actual implementation in real-world scenarios.

**Table 1.** Summary of the comparison between different simulators.

| | FlightGear | XPlane | JMavSim | Gazebo | AirSim | UE4Sim |
|---|---|---|---|---|---|---|
| Commercial Free | Free | Commercial | Free | Free | Free | Free |
| Vehicles | Airplanes | AIrplanes | Multirotor | Multirotor and robots | Multirotor | Multirotor, cars |
| Scene Fidelity | High | Medium | Low | Low | Low | Low |
| Interface ROS | No | No | Yes | Yes | No | No |
| Sensors | Diversity of sensors | Easy incorporation of sensors | No incorporation of sensors | Easy modification of sensors | Monocular depth cameras | Easy modification of sensors |
| Obstacles | Yes | Yes | No | Yes | Yes | Yes |
| SITL-HITL | Yes | Yes | Yes | Yes | Yes | No |
| MAVLink | Yes | Yes | Yes | Yes | Yes | No |
| Ease of Development | Medium | Medium | High | High | Medium | Medium |

Therefore, this paper proposes a method for UAV flight path planning and simulation, specifically dedicated to landslide monitoring. The approach involves constructing a virtual landslide scenario in the Unreal Engine 4 (UE4) and integrating AirSim to simulate UAV flights and reconstruct the landslide area from a 2D to a highly realistic 3D scene. The effectiveness, authenticity, and applicability of the simulation are assessed and the correctness of its flight path planning algorithm is validated by evaluating the quality of the reconstructed model. The main advantage of this method is its ability to significantly reduce the testing costs and risks associated with conducting actual UAV flights in landslide-prone areas. Instead, the proposed simulation framework offers a safe and cost-effective alternative for verifying the accuracy and reliability of the flight path planning algorithm. By providing a realistic representation of landslide scenarios, this approach offers valuable insights for effective disaster prevention and mitigation strategies in areas susceptible to landslides.

The main contributions of this study are as follows:

1. An UAV flight simulation approach that amalgamates UE4 and AirSim is proposed, which address challenges encountered in other simulation frameworks, including texture fidelity, asset constraints, and protocol compatibility and so on;
2. Utilizing the simulation framework to optimize the flight path algorithms, substantiating the practical utility of the proposed framework, and validating the correction of the algorithms;

3. Simulation technology is used in advance to simulate the actual flight, which effectively reduces the input of manpower and material costs, avoids risks, and improves the execution efficiency of actual flight tasks.

## 2. Methods

### 2.1. Landslide Terrain Modeling

2.1.1. Landslide Terrain Modeling

In the simulation environment, the initial step is to build a high-accuracy scene model of the terrain. Advanced UAV technology is employed to capture images of the landslide scene from multiple angles. These images are subsequently utilized for automated matching and solving of aerial triangulation [26], leading to the generation of sparse point clouds based on real images. Further, dense point clouds are reconstructed, and texture mapping is applied to assign high-resolution textures to the 3D model. As a result of this intricate process, a highly realistic 3D scene model is obtained [27].

Aerial triangulation plays a pivotal role in the process of oblique photogrammetry. Its principle is to compute the connection points obtained from the feature matching of images. Subsequently, the coordinates of ground control points measured in reality are employed to incorporate all image areas into the ground coordinate system, yielding the exterior orientation elements for each image. The accuracy of aerial triangulation directly determines the quality of the final results. The fundamental formula for the collinearity condition equation is as follows:

$$x = -f\frac{a_1(X-X_s)+b_1(Y-Y_s)+c_1(Z-Z_s)}{a_3(X-X_s)+b_3(Y-Y_s)+c_3(Z-Z_s)} \quad (1)$$

$$y = -f\frac{a_2(X-X_s)+b_2(Y-Y_s)+c_2(Z-Z_s)}{a_3(X-X_s)+b_3(Y-Y_s)+c_3(Z-Z_s)} \quad (2)$$

Here, $x$ and $y$ represent the image plane coordinates of the image points and $f$ represents camera's focal length. $X_S$, $Y_S$ and $Z_S$ are the object space coordinates of the shooting station. $X$, $Y$ and $Z$ are object space coordinates of object square points. The notations $a_i$, $b_i$, $c_i$ (= 1, 2, 3) comprise the nine directional cosines composed of the three external azimuth elements of the image.

After acquiring the position and orientation of each photo, a sparse point cloud model is generated. To reference the sparse point cloud to the local coordinate system, a height field calculation method based on pairwise depth maps is employed, facilitating the creation of a dense point cloud and mesh. Subsequently, the constructed mesh is overlaid with texture mapping, resulting in the production of a comprehensive 3D model.

2.1.2. Levels of Detail

The rendering capability of modern computer hardware still faces challenges when handling massive terrain datasets, impeding the attainment of real-time performance and higher resolutions [28]. To address this issue, the levels of detail (LOD) technique presents an effective solution. This approach represents terrain with varying levels of detail, utilizing higher detailed models for areas close to the viewpoint and employing coarser terrain models for distant areas, resulting in a visual representation aligning with the capabilities of the human visual system.

The terrain data is stored in the computer as digital elevation model (DEM) data, and it is managed and organized using a pyramid model as shown in Figure 1. The entire area of the landslide terrain is initially structured as a quadtree, with each terrain data block representing a quadtree node [29]. Starting from the root node of the quadtree, each node is recursively subdivided to generate four children nodes at the next level, continuing until the highest resolution of the raw landslide terrain data is reached.

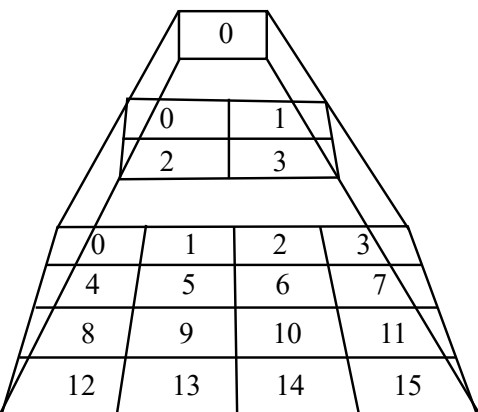

**Figure 1.** Multiscale terrain pyramid.

To simplify the terrain representation and effectively manage large-scale landslide terrain data, the method proposed in this paper adopts a quadtree-based LOD algorithm [30]. This approach organizes and manages the terrain data using the quadtree structure, enabling the selective loading and combination of different resolution levels of landslide terrain data into memory before rendering the scene. The quadtree-based LOD algorithm involves traversing the terrain nodes and determining whether they require subdivision based on certain conditions. Through selective loading and rendering of only the essential portions of the terrain data, the algorithm optimizes memory usage and achieves efficient rendering.

A roughness factor is utilized to account for the required rendering resolution. This roughness factor $R$ represents the steepness of the terrain, with higher values indicating steeper terrain regions that require higher resolution rendering. The mathematical expression for roughness $R$ is as follows:

$$R = \frac{max(\Delta h_1, \Delta h_2, \cdots, \Delta h_6)}{D} \tag{3}$$

Here, $\Delta h_1$ to $\Delta h_6$ denote the differences between the average elevation values at the midpoints of each terrain node's edges and the elevation values at the boundary endpoints, while $D$ represents the edge length of the terrain node. The corresponding terrain representation is illustrated in Figure 2. By calculating the roughness for different terrain regions, the algorithm intelligently selects the appropriate level of detail for rendering, thereby facilitating efficient and realistic visualization of the landslide terrain data.

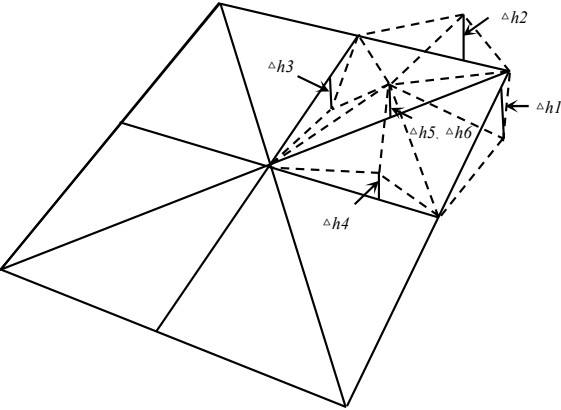

**Figure 2.** Terrain roughness.

## 2.2. Simulation Framework

The simulation framework is built by combining Unreal Engine with AirSim, where Unreal Engine provides highly realistic environment rendering models, and AirSim offers comprehensive control application programming interfaces (APIs), binary interfaces (ABIs), and various sensor models. The confluence of these two constituents enables the creation of a reliable and authentic simulation environment for UAV flight path planning.

### 2.2.1. Unreal Engine

Unreal Engine is a cross-framework game engine developed by Epic Games. As simulation applications have been evolving. Unreal Engine has become a versatile tool used for developing virtual reality and augmented reality applications, movie special effects, industrial simulations, and more [31]. One notable research focus within Unreal Engine pertains to UAV flight path planning. Within Unreal Engine, numerous features are useful for UAV flight path planning. The terrain editor allows the creation of realistic terrains and scenes, further fostering the realism essential to successful simulations. Additionally, UE4 provides extensible plugins, such as virtual reality and networking plugins, which offer further functionalities and support for UAV flight path planning. Unreal Engine also boasts a powerful material editor, enabling the rendering of highly realistic textures, whose lighting, blueprint, and physics rendering features can be employed to implement various interaction and simulation designs, assisting in the creation, testing, and optimization of UAV flight path planning algorithms to meet developers' needs. Furthermore, Unreal Engine's cross-framework advantage allows it to run on various hardware and operating systems, providing a wide range of possibilities for UAV flight path planning applications. Utilizing Unreal Engine to create realistic environments and scenes, automate UAV control, and conduct UAV path planning aids in advancing UAV technology.

### 2.2.2. AirSim

A UAV simulator must create a suitable environment for UAV flight, facilitating the integration of various sensors, such as cameras, LiDAR, GPS, microphones, gas sensors, etc. It also necessitates the incorporation of physical aspects like accurate turbulence, air density, wind shear, clouds, precipitation, and other fluid dynamics constraints. Microsoft's AirSim is a high-fidelity virtual environment based on Unreal Engine that meets these requirements [32]. As an open-source simulation framework developed on Unreal Engine [33], AirSim aims to bridge the gap between simulation and reality, supporting the simulation of diverse vehicles, including UAVs and cars, thereby providing users with a remarkably realistic and immersive experience.

AirSim offers a wide array of callable APIs and ABI interfaces. Throughout the simulation process, AirSim supplies sensor information from the simulated environment to the UAV flight controller. This flight controller then processes the received data using appropriate algorithms and generates control signals for the UAV model, effectively implementing a complete UAV flight design. The API design of AirSim isolates the UAV and flight controller, thereby separating the mathematical model of the simulated UAV from the processing executed by the flight controller. Such design fosters a fully realistic simulation of real-world UAV flights, significantly enhancing the overall realism and effectiveness of the simulation. Additionally, AirSim provides the capability of interfacing with Mavlink, allowing users to employ Pixhawk firmware (e.g., Ardupilot and PX4) for software-in-the-loop (SITL) and hardware-in-the-loop (HITL) simulations. Figure 3 is an example of simulating a virtual drone flight example in the simulation framework.

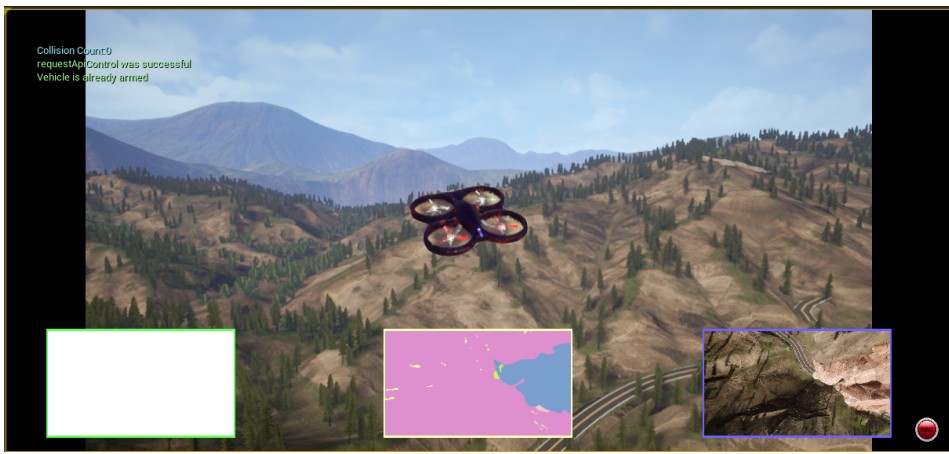

**Figure 3.** An example of virtual drone flight simulation within the proposed simulation framework (windows below, from left to right, show the depth map, semantic segmentation map, and photographic image).

### 2.2.3. Texture Mapping

When rendering large-scale terrain scenes, the use of geometric models to represent natural elements such as grass, soil, and rocks can result in increased complexity in the terrain. To reduce the demands on memory and rendering time while simultaneously enhancing the scene's realism, texture mapping is employed to improve the rendering effect of images [34].

Textures are formed by arranging simple matrices of color data and brightness data [35, 36]. The color of the landslide model's texture is determined by the diffuse reflection and ambient color of the terrain surface, following is the formula, where $C$ represents the color of the landslide model's surface, $C_{ambient}$ is the ambient light color, and $C_{diffuse}$ is the color of the diffuse reflection on the landslide model's surface:

$$C = C_{ambient} + C_{diffuse} \tag{4}$$

Texture mapping refers to establishing a correspondence between the geometric information of objects and texture information based on the spatial position data of the triangle mesh in the model. This process segments the optimal texture mapping for each triangular face, achieving automatic texture mapping of the model and enhancing the visual realism of rendering without adding geometric details to the objects. In texture mapping, $(u, v)$ represents a point in the texture space of the image, and $(x, y)$ are coordinates in the texture image, $(w_{texture}, h_{texture})$ represent the width and height of the texture image, respectively. The formula is described as follows:

$$(u, v) = \left( \frac{x}{w_{texture}}, \frac{y}{h_{texture}} \right) \tag{5}$$

### 2.2.4. Normal Mapping

The level of surface detail depends on how light rays interact with the surface. To achieve a more realistic appearance, it is essential to simulate the irregularities present in real surfaces. The influence of wrinkles on the perception of surface irregularities primarily stems from their impact on the surface's normal direction. To enhance the 3D sense of the object's surface, this paper utilizes normal maps [37] to incorporate bump detail texture information into the material. The height of each texel is encoded using a height field, from which normals are obtained via normalization. Assuming Hg represents the height of the

texel, Ha is the height above the texel, and Hr is the height to the right of the texel, the normal formula is as follows:

$$n = \frac{(H_g - H_r, H_g - H_a, 1)}{\sqrt{(H_g - H_r)^2 + (H_g - H_a)^2 + 1}} \tag{6}$$

Subsequently, the *X*, *Y*, and *Z* components of the normal are stored in the color channels *R*, *G*, and *B*, respectively. However, since normal vectors have signed values within the range of [−1, 1], while the RGB range in the texture is [0, 1] without signed values, it is necessary to compress the normal vector's range. This is accomplished by employing col to represent the compressed normal vector, and its formula is as follows:

$$col = 0.5 \times n + 0.5 \tag{7}$$

After applying the normal map to the texture, when light illuminates the object's surface, different normal directions will present various visual effects.

2.2.5. Lighting

In most cases, visible light illumination models only focus on considering the reflection phenomena of materials. Various simplified models in computer graphics, such as the Phong model [38], Blinn–Phong model [39], and Cook–Torrance model [40], are all based on the global visible light illumination model.

The Blinn–Phong model represents an advanced version of the Phong model, proposed by Blinn, which optimizes and enhances the traditional Phong lighting model. It incorporates the diffuse component of the Lambert model and standard specular mapping. In rendering, the Blinn–Phong model produces smoother and softer highlights compared to the Phong model, with faster calculation speed and heightened realism. Simplifying the incident light into three types: ambient light, diffuse reflection, and specular reflection [41], the Blinn–Phong lighting model is particularly effective in scenarios with a single light source. The calculation formula for the Blinn–Phong model, in such cases, is as follows:

$$I = I_{ambdiff} + I_{ldiff} + I_{spec} = k_a I_a + k_d I_l \cos \theta + k_s I_l (N \cdot H)^{n_s} \tag{8}$$

$$H = \frac{L + V}{2} \tag{9}$$

where *I* is the intensity of the reflected light, $I_{ambdiff}$ is the intensity of the interaction between the diffuse material and the ambient light, $I_{diff}$ is the intensity of the interaction between the diffuse material and the directional light, and $I_{spec}$ is the intensity of the interaction between the specular material and the directional light. $k_a$, $k_d$, and $k_s$ (0 < k < 1) are the reflection coefficients for ambient light, diffuse reflection, and specular reflection, respectively. $I_a$ represents the intensity of the point light source, $\theta$ represents the angle between the incident light direction and the vertex normal, *n* represents the specular exponent, and *N* represents the unit normal vector at the incident point. *H* denotes the intermediate vector between the light direction *L* and the view direction *V*.

2.2.6. Unreal Engine UAV Flight Simulation

Unreal Engine can provide a virtual environment for simulating UAVs' process. In this study, we initially utilized UE4 to simulate the topographical environment of the landslide (Section 2.1) and configure prebuilt scenes. Visualizing the motion of UAVs required adjustments to texture mapping materials and lighting to ensure a realistic simulation. It was essential to establish seamless communication between the constructed landslide model and the 3D environment. Following this, AirSim utilized modules designed for modeling camera sensors. Virtual UAVs were equipped with cameras and LiDAR sensors, placed within the landslide scenario, and sensor data synthesis or detection was achieved. Subsequently, AirSim's API and ABI interfaces were called to provide simulated environ-

mental sensor information to the UAV flight controller during the simulation process. The flight control layer then generated control signals for the UAV model. And the obtained model was stored in the flight system for subsequent route planning and acquisition of high-quality images from the simulated environment (Section 2.3).

### 2.3. UAV Flight Route Planning

2.3.1. Traditional Flight Route Planning

At present, there are various common UAV route planning schemes, such as single horizontal, vertical, and zigzag patterns [42]. These planning schemes can be applied to different terrain conditions and flight missions to meet various task requirements. Single horizontal and vertical line layouts are relatively straightforward but come with certain drawbacks, such as inadequate coverage and overlap, which can affect the quality and accuracy of map data. In some cases, diagonal layouts are necessary to accommodate irregularly shaped areas or complex terrain. The crisscross pattern involves laying out flight paths in a grid-like manner across the target area, allowing the UAV to fly sequentially over each intersection point to complete the entire area's mission. The crisscross patterns can be further refined based on the first two layouts to ensure both heading and lateral overlap, thereby enhancing image quality and effectively avoiding omissions and duplicate captures, thereby improving the efficiency and accuracy of the mission. However, it remains challenging to apply this method flexibly in complex terrain conditions. Figure 4 shows different kinds of traditional flight path planning methods.

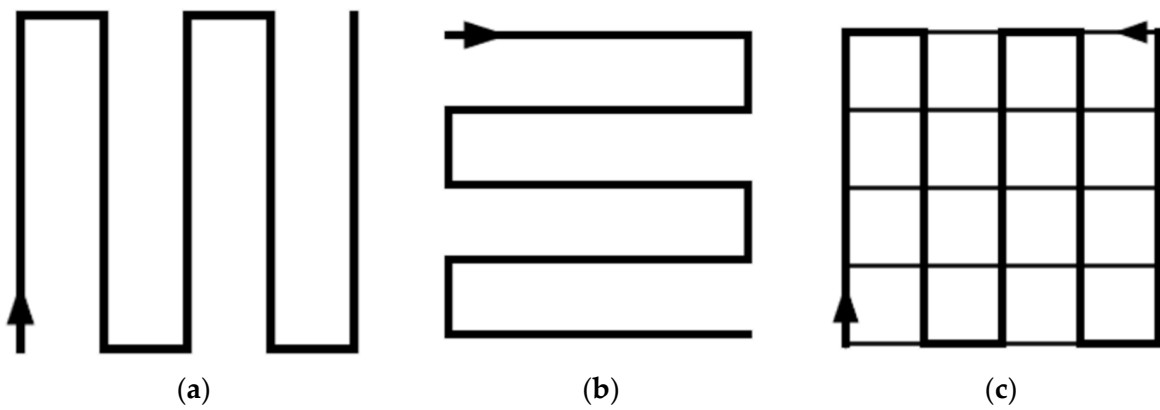

(a)                                        (b)                                        (c)

**Figure 4.** Traditional flight route planning methods: (**a**) single horizontal flight route; (**b**) vertical flight route; (**c**) grid or zigzag pattern flight route.

2.3.2. Novel Route Planning

Traditional UAV route planning methods fall short in capturing the vertical details of landslides and encounter issues such as the failure of 3D densification computation. In this study, a novel terrain-following flight approach is proposed. It necessitates continuous images with a specific ratio of heading and lateral overlap during the flight, and it demands simultaneous image capture of the landslide area to obtain higher-quality 3D textures. Otherwise, unrelated shadows or reduced visual disparities could degrade the model reconstruction quality. The method first utilizes UAV oblique photogrammetry to generate a digital elevation model (DEM) of the target area and subsequently generates contour lines with 30 m spacing to depict terrain contours optimally. Using contour lines to design UAV routes allows for better representation of variations in terrain elevation. Contour line density can be controlled by the route density, with higher contour line densities requiring more routes. Typically, DEM-generated contour lines exhibit numerous inflection points and complexity. Planning routes directly based on these contour lines would increase the UAV's energy consumption, thus reducing its flight endurance. Frequent changes in direction could also adversely affect flight safety. In practice, both flight endurance and safety are critical factors for UAVs. Therefore, contour lines must be smoothed to

better support route planning and flights. This study employs the Douglas–Peucker algorithm [43] to smooth the contour lines, reducing the number of inflection points, lowering computational complexity, and enhancing route planning efficiency. Subsequently, contour lines are stretched based on factors such as object heights and UAV flight altitudes to achieve the conversion from contour lines to UAV routes. Figure 5 shows the key steps of the novel route planning method.

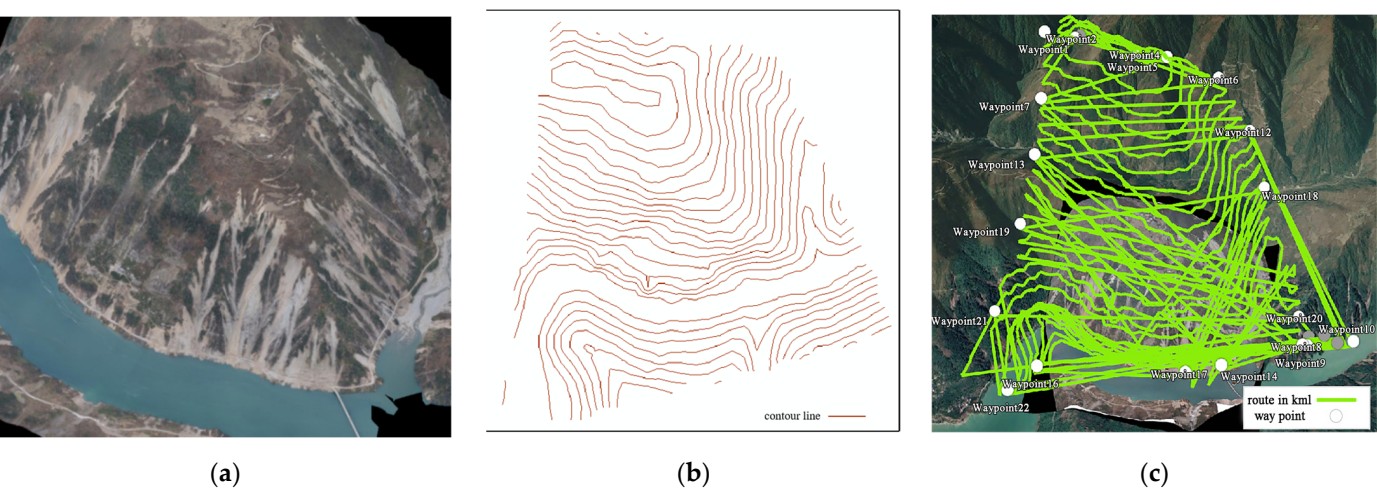

(**a**)  (**b**)  (**c**)

**Figure 5.** The main steps of the novel route planning method: (**a**) orthorectified image of the survey area; (**b**) contour map (in SHP format); (**c**) UAV flight route (in KML format).

## 3. Experiment and Results

To ensure the visual realism of the experimental scene, the safety of UAV flight, and to test the accuracy of the flight path planning method, this experiment combines the power of Unreal Engine and AirSim for validation. Firstly, landslide datasets are acquired and utilized for 3D reconstruction, creating a simulated terrain model. Subsequently, Unreal Engine is leveraged to fine tune the lighting and textures, enhancing the simulated environment to closely resemble reality, and AirSim is employed for UAV flight path simulation. Ultimately, validating the correctness of the flight path planning algorithm.

### 3.1. Data Set

In this study, two sets of landslide image data were obtained from the SkyPixel website [44]. These images capture two typical landslide incidents in the vicinity of Lianhe Village, Luding County, Ganzi Prefecture, Sichuan Province. Figure 4 shows the original images of the composite and steep types of landslides. The composite landslide is composed of a variety of landslide types such as notch, bulge, and fracture. They change from the slope slide to rock mass slide or collapse. The steep landslides are convex structures with steep slopes. With fast sliding speed, the sliding will be accompanied by large noise and vibration. The original videos possess a frame rate of 30 frames per second with a size of 1920 × 1080 pixels and last for a duration of 25 s. To reduce the processing workload of analyzing all frames in the video, 100 images were extracted from each video at a time interval of 0.25 s. Figure 6 shows the original images of the two types of landslide.

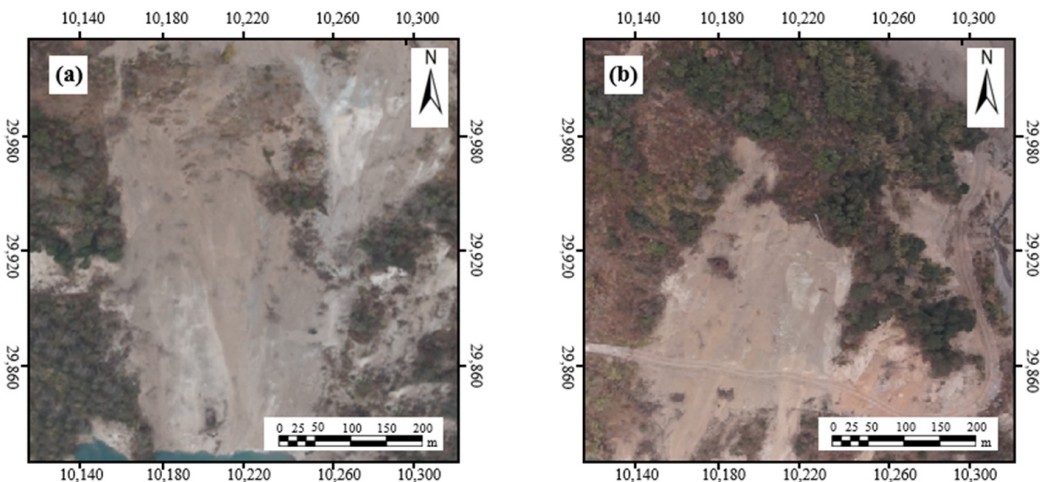

**Figure 6.** The original landslide image: (**a**) compound-type landslide; (**b**) steep-type landslide.

*3.2. Landslide Model Construction*

Firstly, aerial triangulation techniques were used to generate dense point clouds, and irregular triangulated meshes were created based on the point cloud data. Secondly, the texture of the landslide was obtained from the aerial images. However, there were noticeable color differences between the texture maps obtained from the landslide images and the terrain scene texture maps. These differences manifested as higher exposure, brighter colors, lower contrast, and lower saturation. To achieve color consistency between the two texture maps, the color matching method was employed, which successfully aligned the color information of the terrain scene texture map with that of the landslide texture map. Subsequently, the landslide model and its texture map were unified to the WGS-84 coordinate system, and texture mapping was employed to seamlessly apply the texture to the landslide model. Through this process, a final output of a highly realistic 3D model of the landslides was obtained.

To achieve seamless height blending between the landslide model and the terrain scene, as well as to apply the appropriate material to the landslide model in Unreal Engine, a material sphere was meticulously constructed and subsequently assigned to the landslide model. In cases where the model displays pronounced specular highlights under scene lighting, it becomes essential to fine tune the roughness and metallic parameters of the material to minimize the intensity of reflections. The resulting landslide model crafted through this experimental process, is illustrated in Figure 7.

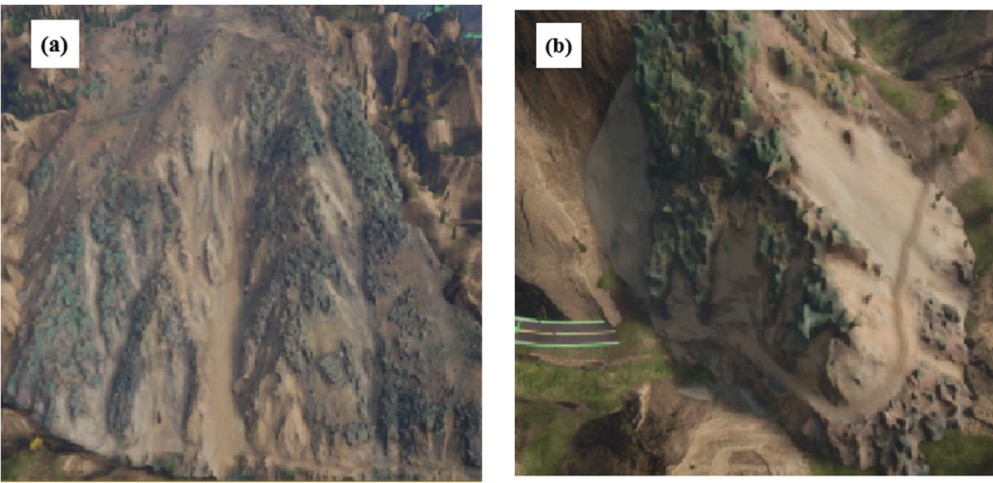

**Figure 7.** Slide model: (**a**) compound-type landslide; (**b**) steep-type landslide.

*3.3. UAV Flight Path Planning Simulation Implementation*

In this study, the flight path algorithm for the compound-type landslide was initially simulated and validated. Following this, a second validation was conducted using the steep-type landslide with the same novel method to ensure the algorithm's applicability across various landslide scenarios and to verify the realism and efficacy of the simulation framework.

Utilizing the previously constructed terrain scene, the UAV flight path planning method was introduced for simulation in this experiment. The UAV was tasked to follow the meticulously planned flight path, capturing image data of the simulated experimental scene during the flight. Subsequently, the captured image data was used for the 3D reconstruction of the simulated terrain scene. Firstly, flight paths were meticulously planned for both the composite-type and steep-type landslides using both the traditional horizontal flight surface mode and the novel terrain-following method proposed in this paper. Subsequently, the UAV followed the predefined flight paths and systematically collected images of the experimental scene at regular time intervals within the simulation framework to obtain DEM data of the landslide area.

During the image collection process, ensuring consistent heading overlap was of paramount importance, and the camera tilt angle was set to be vertically downward. The relevant parameters for UAV flight were thoughtfully configured, as illustrated in Table 2.

**Table 2.** Flight parameter setting.

| Type of Landslide | Flying Height/m | Route Mode | Flight Runtime/s | Time Interval/s | Flight Distance/m | Ground Resolution/cm/px | Number of Images/Sheet |
|---|---|---|---|---|---|---|---|
| Compound type | 0–360 | traditional method | 256 | 2 | 1024 | 10.6 | 128 |
| | | Novel method | 250 | | 1024 | 8 | 125 |
| Steep type | 0–200 | traditional method | 264 | 3 | 1024 | 11.92 | 88 |
| | | Novel method | 255 | | 1024 | 7 | 85 |

The obtained image data was processed to generate a real point cloud based on the actual images. Subsequently, a dense reconstruction process was employed to produce a high-resolution dense point cloud complemented by authentic textures, resulting in a 3D model with high-resolution and authentic textures. To evaluate the models reconstructed using the two flight path planning methods, a thorough examination of texture details was conducted. This analysis served to validate the advantages of the novel terrain-following flight path method and assess the overall quality of the simulation framework. Figure 8 presents the results of the 3D reconstruction using different methods.

*3.4. Result*

3.4.1. 3D Model Completeness

In this experiment, we initially evaluate the merits of the two different methods in a qualitative manner, and meticulously compared the 3D models of the landslides generated by the two different flight path planning methods. The evaluation focused on two crucial aspects: overall completeness and local feature details.

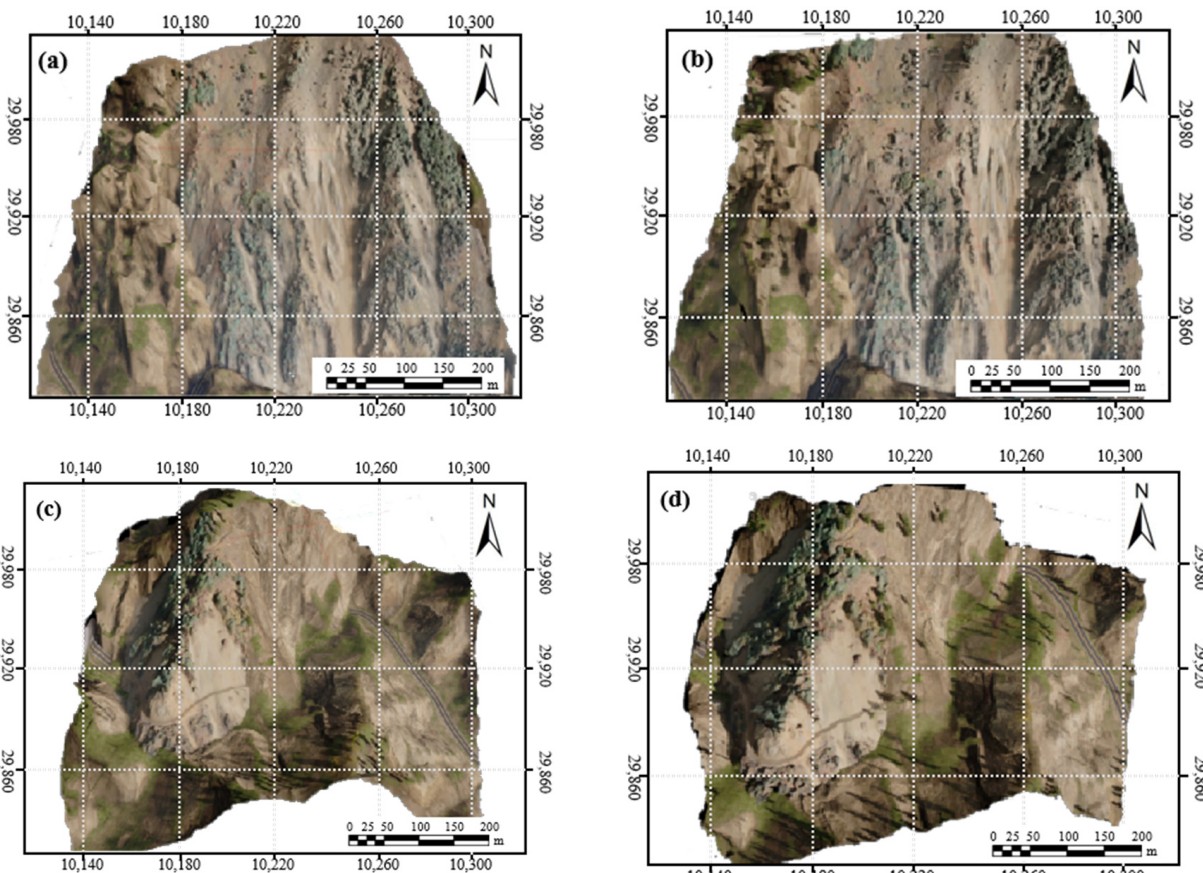

**Figure 8.** 3D reconstruction model: (**a**) model of compound-type landslide with traditional method; (**b**) model of compound-type landslide with novel method; (**c**) model of steed-type landslide with traditional method; (**d**) model of steed-type landslide with novel method.

In terms of overall completeness, both flight path planning methods accurately re-produced the overall shape of the landslides, without any significant details missing or shape distortions observed in either case. The models exhibited a faithful representation of the landslide topography, validating the effectiveness of both methods in capturing the general shape of the terrain. However, when evaluating local feature details, noteworthy differences emerged between the two approaches. The traditional terrain-following flight path method often resulted in slightly blurry ground information and exhibited some deformation in vegetation. In contrast, the 3D reconstruction results using the novel terrain-following flight path design method showed higher overall resolution and clearer ground details. The novel method remarkably captured vegetation coverage and rock distribution characteristics with exceptional fidelity, revealing minimal deformation in vegetation and rocks. These observations demonstrate the superior model reconstruction effects offered by the novel terrain-following flight path method. Figure 9 represents the two types of landslide local details.

The experiment highlights the advantages of the proposed terrain-following flight path design method in attaining higher-quality 3D reconstruction results, effectively capturing intricate local features and producing more realistic and visually appealing models of the landslides.

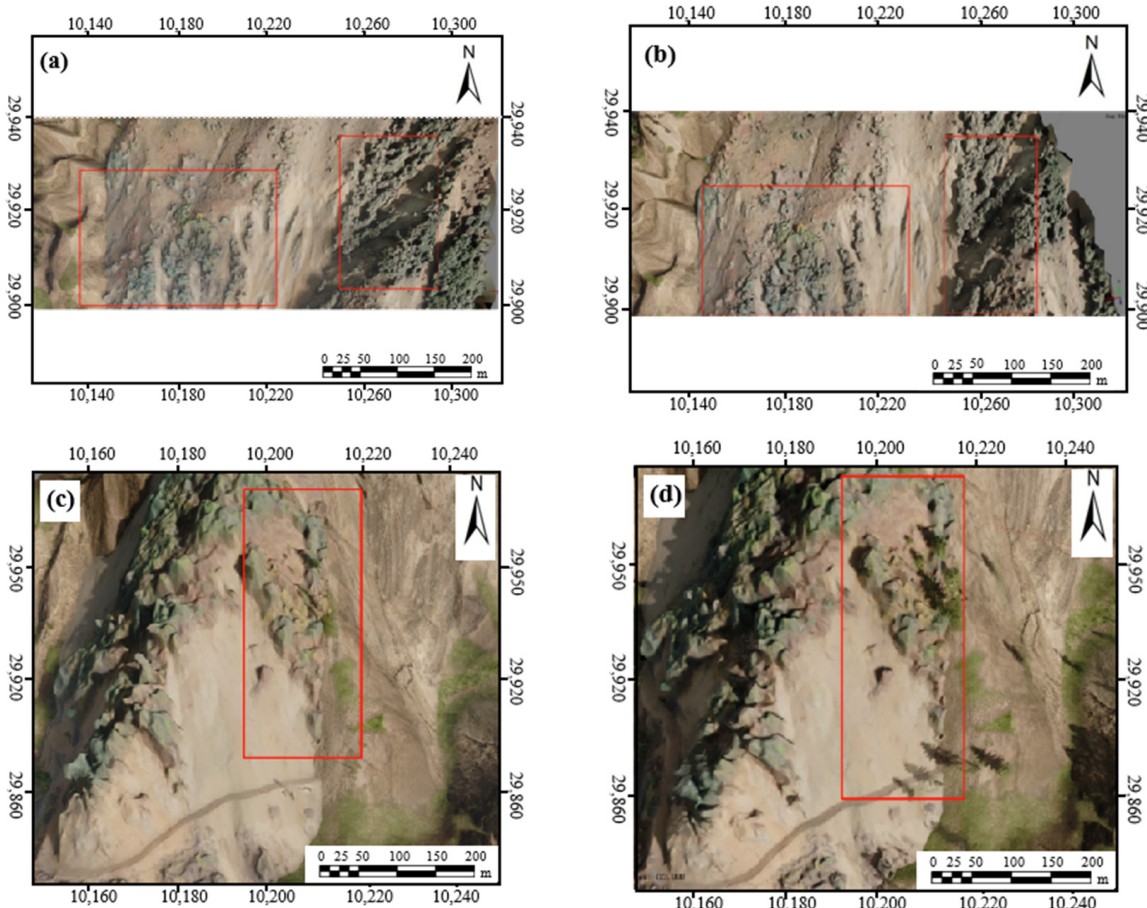

**Figure 9.** Landslide local details: (**a**) traditional method used in compound-type landslide; (**b**) novel method used in compound-type landslide; (**c**) traditional method used in steed-type landslide; (**d**) novel method used in steed-type landslide.

### 3.4.2. 3D Model Accuracy

Subsequently, we quantitatively compare the two different methods. We selected the RMSE as the analytical indicator. The most crucial aspect of the landslide model is to ensure its geometric accuracy while reproducing the real form. In this experiment, the length and width information of the landslides obtained from the image maps were taken as the actual values, while the corresponding geometric length values on the 3D models were considered as the measured values. By computing the root-mean-square error, the experiment was able to quantify the level of accuracy achieved by the 3D models in replicating the geometric attributes of the landslides, providing a quantitative measure of the model's fidelity to the real-world topography. The formula is as follows:

$$RMSE = \sqrt{\frac{\sum_{i=1}^{n}(l'-l)^2}{n}} \tag{10}$$

where *RMSE* represents the root-mean-square Error, l denotes the actual value, and $l'$ represents the measured value, *n* represents the number of measured edges.

The calculation results of the geometric length RMSE for different flight path planning methods are presented in Table 3. The table reveals that the 3D model accuracy of both methods ranging from 10 to 14 cm. Specifically, the traditional flight path approach results in model accuracy between 12 and 14 cm, while the novel flight path method improves model accuracy to the range of 10 to 11 cm. Notably, the 3D models generated using the novel terrain-following flight path method demonstrated significantly higher precision,

with the novel method outperforming the traditional one by 2 to 3 cm in precision. Among them, the compound-type landslide model reconstructed using the novel terrain-following flight path algorithm achieved the highest accuracy, with an RMSE of 10.807 cm. Conversely, the 3D models generated using the traditional flight path planning approach displayed noticeable issues, including overall distortion, loss of texture, and displacement of details. As a result, these models exhibited larger measured errors in geometric length, indicating a comparatively lower level of accuracy in replicating the real-world topography.

**Table 3.** RMSE of 3D models generated via different routes.

| Type of Landslide | Route Type | RMSE/cm | |
|---|---|---|---|
| Compound type | Traditional method | 12.457 |  |
| | Novel method | 10.807 | |
| Steep type | Traditional method | 13.684 | |
| | Novel method | 11.121 | |

The observed improvement in RMSE is statistically significant, indicating the superiority of the proposed flight path approach in terms of monitoring precision. In the context of geological hazard monitoring, even minor improvements can have a crucial impact on accuracy, enabling monitoring agencies to better predict and respond to geological disasters. The refined accuracy offered by the novel flight path method is paramount in enhancing the overall efficacy of geological hazard monitoring efforts. Its 2 to 3 cm accuracy improvement is important for making quick decisions and taking action in emergency situations. Furthermore, the quantitative analysis using RMSE as the evaluation metric provides a robust foundation for assessing the performance of the two methods. The narrower range and lower values of RMSE associated with the novel flight path approach highlight its superiority in capturing the intricacies of the terrain and geological features. This reduction in RMSE reflects a more faithful representation of the actual landscape, emphasizing the method's capability to produce highly accurate three-dimensional models.

The results above clearly demonstrate the substantial improvement achieved by the novel terrain-following flight path method in terms of geometric precision, making contributions to attain more reliable and precise 3D reconstructions of landslide scenarios.

Additionally, the novel terrain-following flight path algorithm was found to reduce the total flight time by 9.3% compared to the traditional algorithm, which partly alleviates the issue of UAV endurance, allowing for more efficient and extended data collection periods during flight operations. Moreover, the simulation results for the steep-type landslide scene were found to be in line with those of the composite-type landslide scene. This observation underscores the versatility and applicability of the method employed in this study, demonstrating its effectiveness in addressing different landslide scenarios.

In conclusion, the combination of improved flight efficiency and consistent performance across diverse landslide scenarios highlights the potential of the novel terrain-following flight path algorithm as a valuable tool for UAV-based terrain data acquisition and 3D modeling in landslide monitoring and analysis applications, simultaneously confirming the practicability and authenticity of the simulation framework proposed in this paper.

## 4. Discussion

In this study, we propose a low-cost UAV flight path validation method based on Unreal Engine and AirSim. This method allows for the testing of various flight path algorithms within a preconstructed landslide simulation scene, ultimately resulting in safe flight paths for efficiently acquiring the necessary imagery for 3D modeling in complex environments. Compared to conducting UAV flights directly in the field with unvalidated flight paths, this approach offers high fault tolerance and cost effectiveness and significantly reduces the risk of UAV damage due to the complexity of the landslide scenario or potential parameter errors.

In contrast to some other UAV simulation frameworks, Unreal Engine stands out for its capability to create complex and highly realistic scenes, providing cross-framework compatibility across various hardware and operating systems. Combined with AirSim, it enables the creation of UAV motion planning with support for motion capture, providing the information caught by the sensors to UAV flight controllers obtained from the simulation environment, ensuring the realism and effectiveness of the simulation. In the landslide 3D reconstruction experiments, this study utilized two sets of typical landslide images from the vicinity of Lianhe Village, Luding County, Ganzi Prefecture, Sichuan Province, to construct high-accuracy compound-type and steep-type landslide scenes, meeting the requirements for subsequent UAV flight path planning simulations.

In the UAV flight path planning simulation experiments, flight path validation was conducted for both types of landslide scenarios, successfully demonstrating the applicability of our proposed method to different landslide scenes. Traditional contour-based UAV flight path planning leads to frequent sharp turns. To reduce computational complexity and enhance UAV flight efficiency, we applied the Douglas–Peucker algorithm to smooth the contour lines and stretch them to optimize flight paths within the landslide scenarios. The results indicate that both flight path design solutions are safe and accurately reproduce the overall morphology of the landslides. Compared to traditional flight path planning methods, the new approach is better suited for landslide terrain, resulting in higher-resolution and clearer ground detail models. Additionally, the new flight path requires less time by 9.3%, resulting in higher overall 3D model accuracy.

Apart from the studies discussed above, the safety of UAV is a critical issue. Prior studies have noted the importance of drone route safety. In the simulation framework of this study, adaptive adjustments can be made to UAV parameters, camera parameters, and the quality of generated images. Factors such as the flight speed of the UAV, safe clearance from on-site obstacles, and minimum altitude above ground can all be considered in the virtual scene. Due to the limited battery capacity of UAVs, which decreases with battery aging, the flight duration of UAVs is restricted.

Furthermore, the proposed method allows for further adjustments to the flight time and path, placing suitable image acquisition viewpoints at key locations to enhance the quality of model reconstruction. It can avoid occlusion or collision caused by on-site obstacles, guaranteeing the safety of the UAV and the quality of the captured images. Importing the optimized flight path parameters into the UAV effectively reduces costs and risks during actual flights, improving landslide monitoring efficiency and accuracy.

## 5. Conclusions

This study aimed to address the challenges and complexities associated with UAV flights in landslide areas. By integrating Unreal Engine and AirSim, the researchers successfully implemented UAV flight path planning and simulation. The simulation framework allowed for the creation of a realistic landslide model and enabled the design and execution of UAV flight paths within the simulated environment. The captured images during the UAV flight simulation were then utilized for precise 3D reconstruction, and the reconstructed models were evaluated based on their texture details. The experimental results demonstrated that the simulation framework developed in this study provided a favorable environment for conducting UAV flight path planning and simulation experiments.

Additionally, the proposed novel terrain-following UAV flight path planning method out-performed the traditional approach, resulting in finer texture details with an accuracy of 10.807 cm, which affirms the correctness of the flight path algorithm. The establishment of the UAV flight path simulation framework contributes to timely and efficient emergency response in the event of landslide disasters. Furthermore, the validated flight path algorithm ensures accurate flight path planning, leading to reduced costs and risks during real UAV flights, and enhances the efficiency of landslide monitoring. The application of this framework and validated flight path algorithm in natural disaster scenarios promises to make a valuable impact in mitigating and managing landslide events.

**Author Contributions:** Conceptualization, D.X., R.H. and C.W.; methodology, D.X., R.H. and C.W.; software, D.X. and R.H.; investigation, D.X. and R.H.; formal analysis, D.X. and R.H.; writing—original draft preparation, D.X. and R.H.; and writing—review and editing, D.X., R.H., C.W., C.Z., H.X. and Q.L. All authors have read and agreed to the published version of the manuscript.

**Funding:** The work is jointly supported by the Chang'an University (Xi'an, China) through the National Key Research and Development Program of China (Grant No. 2020YFC1512001), the Shenzhen Scientific Research and the Development Funding Programmes (Grant No. RCYX20210706092140076), the Open Research Fund from Guangdong Laboratory of Artificial Intelligence and Digital Economy (SZ) (Grant No. GML-KF-22-15), the National Natural Science Foundation of China (Grant No. 42374018) and Open Research Fund of State Key Laboratory of Subtropical Building and Urban Science (Grant No. 2022ZB04).

**Data Availability Statement:** The data generated and analyzed during this study are available from the corresponding author on request.

**Conflicts of Interest:** The authors declare no conflict of interest.

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
