# Peer review of "A Simulation Framework of Unmanned Aerial Vehicles Route Planning Design and Validation for Landslide Monitoring"

_remotesensing, doi:10.3390/rs15245758_

Round 1

Reviewer 1 Report

Comments and Suggestions for Authors

Dear Authors,

I have read the paper carefully since the topic has caught my attention. The initial vision of the paper that came to me from the reading of the abstract, unfortunately, did not meet my expectations after reading the whole article.

The idea itself is good, but the article has many flaws and needs to be investigated better, and the results need to be presented better, in a more scientific way, overall it needs more scientific soundness.

The whole novel method is not well presented to the readers. I.e. too much space has been taken by the chapters 2.2.3 to 2.2.6. but with small merit and impact on route planning and project design.

The difference between the novel and the traditional method is not well described.

The figures are of poor quality and give little information and clarification ("the figure should be worth 1000 words"); there is no scale or coordinate grid, especially Fig. 3 and all other figures in relation to Fig. 3. There is no flight path algorithm defined or shown in figures. Table 1 showing flight parameter settings lacks additional info i.e. flight length...  Table 1 should be accompanied by a figure showing flights.

Figures 7 and 8 showing differences between the novel and the traditional method are in different times of day - the shadows are different, and again, there is no scale or any kind of grid.

The differences shown in the paper rely too much on the subjective effect of the visualisation than on the actual numerically provable.

Regarding the 3D model accuracy, why are there no actual field measurements to compare to? 

Line 476 - 478 - bookmark errors

I propose that you rewrite the paper, taking into account the above comments but from the positive side. 

Best regards and good luck

Comments on the Quality of English Language

English language should be checked for clarity and soundness.

Reviewer 2 Report

Comments and Suggestions for Authors

1. There are many missing references. For example, there are equations and statements such as "method X was used for this" that should be backed up by a reference.

2. Unreal Engine can be used within Matlab. See https://www.mathworks.com/help/uav/ug/how-3d-simulation-for-uav-works.html  How was it used?

3. The framework seems to be depend on previously obtained images to generate images.

4. Many frameworks were presented in the paragraph beginning in line 71. How is yours better? The authors should clearly compare the new framework with the ones presented in that paragraph.

5. RMSE gain is marginal, less than 1 dB. Is it enough to change a decision? Sometimes, in an emergency situation, one must consider the tradeoff between time and accuracy. Is the presented framework faster than the traditional methods?

6. Coverage of how Unreal Engine and AirSim were used is superficial. A third party would not be able to replicate the framework with given information

7. The authors should look at https://cdn2.unrealengine.com/unreal-engine-project-antoinette-information-paper-9885c3cfac34.pdf  and https://www.mdpi.com/2072-4292/14/13/3200 , for example.

Comments on the Quality of English Language

Some acronyms not defined, e.g. UE4.

Minor review necessary, in particular using commas instead of dots, some capital letters in the middle of sentences, etc.

Line 476, bookmark not defined.

Reviewer 3 Report

Comments and Suggestions for Authors

1. The abstract should further refine the results and thus be able to emphasize the correctness of the new algorithm.
2. The time saves of 9.3% is obtained with the same flight path?
3. Some errors must be corrected in the introduction, including: the trigger for landslides is not solely gravity, landslides do not necessarily exhibit high frequencies and velocities, and there are numerous low-angle slow slides (line 30); GNSS and inclinometers are equally capable of real-time monitoring (line 48).

4. At line 188, x0,y0 do not appear in equations (1) and (2), or x0,y0 also represent the image plane coordinates of the image points just like x,y.

5. In equation (5), (u,v) represents a point in texture space, is this a plane coordinate? The right-hand side is a spatial coordinate, can the "=" connect these two be directly?

Reviewer 4 Report

Comments and Suggestions for Authors

The authors presents the method in a good overall presentation but have some comments to improve the paper:

1- in the contributions section, points 2 and 3 are similar so how can we differentiate them

2- In the methods section, I prefer to have figures to ease the overall understanding such as Figures that illustrate the simulation framework, unreal engine, and Airsim

3- Can the results presented here compared with any framework in the literature to present the evaluation 

4-I suggest adding some security analysis in the discussion section based on using the proposed method

5- some references are too old, you should use references from a maximum of last five years

Comments on the Quality of English Language

needs some minor checks 

Round 2

Reviewer 3 Report

Comments and Suggestions for Authors

In this paper(remotesensing-2667248), the authors propose the use of the Unreal Engine simulation framework to design UAV flight path planning specifically for landslide monitoring. I found the image results acquired by the authors using the new algorithm to be extremely impressive and useful for monitoring landslides. The authors have already have revised some of the deficiencies in the paper, but still, as with the previous proposal, I do not comprehend the new algorithm all that well. I am unable to determine the accuracy of the new algorithm with precision. I suggest that algorithm-savvy, more specialised reviewers to make the final decision.

Author Response

Dear Reviewer:

We appreciate your positive and constructive comments and suggestions on our manuscript. We have studied your comments carefully and have made revisions, which are marked in blue in the paper, mainly located in line 519. Attached, please find the revised version, which we would like to submit for your kind consideration.

First of all, given your limited comprehension of the new algorithm, we'll summarize the content and general process in simple terms. The new method addresses the shortcomings of traditional approaches to capturing texture details during image acquisition. The process is as follows: Firstly, obtain the Digital Elevation Model (DEM) of the landslide area and generate contours. Due to the complexity of contours introducing negative impacts on drone flights, the Douglas-Peucker algorithm is employed to smooth the contours, resulting in the final drone flight path.

Furthermore, in response to your comment on the inability to precisely judge the accuracy of the new algorithm, we would like to clarify: Firstly, our method exhibits higher precision compared to traditional approaches. Secondly, in geological hazard monitoring, even minor improvements can have a crucial impact on accuracy. Lastly, the new method demonstrates a 9.3% improvement in time efficiency. This implies not only enhanced accuracy but also the ability to rapidly deliver results, which is crucial for making prompt decisions and taking action in emergency situations.

All above is my further discussion and clarification on the methods and results, which I hope will help you understand the novel method.

Looking forward to hearing from you.

Thank you and best regards.